# Changes in the gut microbiota of Nigerian infants within the first year of life

Omolanke T. Oyedemi[1], Sophie Shaw[2], Jennifer C. Martin[3], Funmilola A. Ayeni[1,4]*, Karen P. Scott[3]*

1 Department of Pharmaceutical Microbiology, Faculty of Pharmacy, University of Ibadan, Ibadan, Nigeria, 2 Centre for Genome Enabled Biology and Medicine, University of Aberdeen, Aberdeen, Scotland, United Kingdom, 3 The Rowett Institute, University of Aberdeen, Aberdeen, Scotland, United Kingdom, 4 Department of Biology, Simmons University, Fenway, Boston, United States of America

* k.scott@abdn.ac.uk (KPS); ayeni@simmons.edu (FAA)

## Abstract

The composition of the gut microbiota plays an important role in maintaining the balance between health and disease. However, there is considerably less information on the composition of the gut microbiota of non-Western communities. This study was designed to investigate the evolution in the gut microbiota in a cohort of Nigerian infants within the first year of life. Faecal samples were obtained monthly from 28 infants from birth for one year. The infants had been born by a mix of natural birth and caesarean section and were either breast-fed or mixed fed. Sequencing of the V1-V2 region of the 16S rRNA gene was used to characterise the microbiota. Short chain fatty acids and lactate present in each faecal sample were identified by gas chromatography. Microbial differences were observed between the vaginal and caesarean section delivered infants in samples collected within 7 days of life, although these differences were not observed in later samples. Exclusively breastfed infants had predominance of *Ruminococcus gnavus*, *Collinsella*, and *Sutterella* species. Different *Bifidobacterium* species dominated breast-fed compared to mixed fed infants. *Clostridium*, *Enterococcus*, *Roseburia*, and *Coprococcus* species were observed once the infants commenced weaning. Butyrate was first detected when weaning started between months 4–6 in the majority of the infants while total short chain fatty acid concentrations increased, and acetate and lactate remained high following the introduction of solid foods. The observed taxonomic differences in the gut microbiota between Nigerian infants, as well as butyrate production during weaning, were strongly influenced by diet, and not by birthing method. Introduction of local/solid foods encouraged the colonisation and evolution of specific marker organisms associated with carbohydrate metabolism.

## Introduction

Humans play host to a complex microbial community referred to as the human microbiota, with the total number of genes encoded by these microbial genomes, the human microbiome, exceeding the number of human genes. The large intestine harbours the largest number of

**Data Availability Statement:** The 16S rRNA gene sequences were deposited in the European Nucleotide Archive (ENA) under the Accession number PRJEB31073. The remaining relevant data

are within the article and its Supporting information files.

**Funding:** OTO was a self-funded PhD student performing her PhD research at the University of Ibadan, Nigeria. OTO received no specific funding for her PhD work except that her 5 month placement at The Rowett Institute was supported by the Scottish Funding Council - Global Challenge Research Fund (Internal Grant reference SF10180). The Rowett Institute (University of Aberdeen) receives financial support from the Scottish Government Rural and Environmental Sciences and Analytical Services (RESAS). The funders had no role in study design, data collection and analysis, decision to publish, or preparation of the manuscript.

**Competing interests:** The authors have declared that no competing interests exist.

microbes when compared to other human body sites. This microbiota includes bacteria, viruses, eukarya (protozoa, yeast and fungi) and archaea [1, 2].

The gut microbiota plays an important role in shaping the immune system, preventing colonisation by pathogenic microorganisms and improving the health and development of infants. It's multifunctional capacity led to the gut microbiota being defined as a "super organ" [3]. Perturbation in the gut microbiota has been associated with different disease pathologies such as immune diseases like atopy, allergy, asthma, and sclerosis [4]; autoimmune diseases [5]; metabolic diseases, such as diabetes and obesity [6, 7]; and gastrointestinal diseases like diarrhoea, inflammatory bowel diseases, and necrotizing enterocolitis [8].

Short chain fatty acids (SCFAs) are secondary metabolites produced from the fermentation of non-digestible carbohydrates (NDC) by the gut microbiota. These carbohydrates escape degradation by host digestive enzymes and reach the large intestine where the gut microbiota ferments them as substrates producing various metabolic products including the three major SCFAs acetate, butyrate and propionate, and also products like formate, lactate and succinate. The SCFAs are readily taken up by the mucosal epithelial cells in the gastrointestinal tracts and are involved in cellular metabolism such as gene expression, proliferation, chemotaxis, apoptosis and differentiation of immune cells [9].

Infants are generally thought to be born with intestines that are sterile or that contain a very low level of microbes, however, the infant gastrointestinal tract is rapidly colonised following delivery [10, 11]. The initial residents of the gut after birth are usually facultative anaerobes, including Proteobacteria (Enterobacteriaceae) and some Firmicutes (*Staphylococcus*, *Enterococcus*). After the exhaustion of any oxygen by these initial colonising bacteria, the environment is ready for colonisation by the strictly anaerobic bacteria within the Actinobacteria, Bacteroidetes and Firmicutes. Various factors contribute to the order of colonisation and development of the gut microbiota in infants, including: maternal microbiota, gestational age, genetics, delivery method (vaginal/natural birth, caesarean section), diet (breastfed, formula-fed, mixed-fed), and antibiotic use [3, 12]. In general, the infant's gut composition is dominated by the genus *Bifidobacterium* in the first year of life as widely reported within many different studies of Western infant cohorts. A comparative gut microbiome study between Malawian, Amerindian and USA children also showed the dominance of *Bifidobacterium* species across the three populations in infants for the first year [13]. This study showed that the high conservation between the early infant samples in the different populations decreased following weaning, and the overall microbial diversity increased. By two years of age, the microbiota of the three geographically different populations were readily distinguishable [13]. A study comparing the microbiota in children under 6 years old from Europe (Italy) and Africa (Burkina Faso) showed that the main overlap in the microbial populations occurred in the samples from the youngest children (under 2yr) when *Bifidobacterium* species were prevalent. *Bifidobacterium* species became virtually undetectable in the older Burkina Faso children although remained at levels of 2–10% in the Italian children [14]. A longitudinal study carried out on cohorts of Finnish, Russian and Estonian infants over three years showed that the *Bifidobacterium* species detected changed after weaning, with *B. longum* subspecies *infantis* more prevalent during breastfeeding than after weaning [15]. These previous studies into the colonisation and adaptation of the infant gut microbiota are either single time point [13, 16–18] or longitudinal studies [12, 14, 16, 19] within Western communities. There is no study investigating the development of the infant gut microbial composition over time from Nigeria, or any other African country. The present study investigated the changes in the gut microbiota of a cohort of Nigerian infants within the first year of life based on delivery mode, age and diet.

## Materials and methods

### Study subjects

This longitudinal cohort study of sampled participants was approved by the Ethics and Research Committee of the Federal Teaching Hospital, Ido-Ekiti, Ekiti State, South/West Nigeria, with protocol number: ERC/2016/09/29/44B. Written informed consent was obtained from the infant's parents. Twenty eight infants (9 males and 19 females) were recruited, with mean gestational age (37.6±2.8 weeks) and birth weight (2.9±0.6 kg). Inclusion criteria were caesarean section and vaginal birth babies, full term and preterm babies, exclusively breastfed and mixed fed (breastfed and formula fed combined) babies and those with and without antibiotic treatment. Babies that were infected with HIV, tuberculosis and pneumonia were excluded from the study. Infants were grouped into two categories for birthing method: caesarean section birth (CSB) or vaginal birth (VB), and four categories for feeding: preweaning samples which split into babies either exclusively breastfed (EBF), or those who were a mixed fed a combination of breast and formula milk (MF), or combined as grouped preweaning samples in the analysis, and the weaning samples where babies were fed a combination of local and solid foods (local/solid fed -LF/SF) (Table 1).

**Table 1. Characteristics of study infants at birth.**

| Baby code | Gestational age (week) | Mode of delivery | Sex | Weight at birth (kg) | Feeding method | Antibiotic treatment |
|---|---|---|---|---|---|---|
| Baby 1 | 37 wks 5 d | CSB | Male | 3.45 | MF | No |
| Baby 2 | 39 wks 5 d | CSB | Female | 3.8 | MF | No |
| Baby 3 | 39 wks 4 d | VB | Female | 3 | MF | No |
| Baby 4 | 38 wk 5 d | VB | Female | 2.8 | EBF | No |
| Baby 5 | 37 wks 3 d | VB | Male | 3.25 | EBF | No |
| Baby 6 | 39 wks 6 d | CSB | Male | 2.9 | EBF | No |
| Baby 7 | 42 wks | CSB | Female | 3.2 | MF | Yes |
| Baby 8 | 38 wks | VB | Female | 2.8 | EBF | Yes |
| Baby 9 | 38 wks 2 d | VB | Female | 3.2 | MF | No |
| Baby 10 | 34 wks 5 d | CSB | Female | 2.5 | MF | No |
| Baby 11 | 34 wks 5 d | CSB | Male | 2.6 | MF | No |
| Baby 12 | 38 wks 5d | CSB | Female | 2.6 | MF | Yes |
| Baby 13 | 38 wks 5d | CSB | Female | 2.4 | MF | Yes |
| Baby 14 | 40 wks 4d | VB | Female | 2.6 | EBF | No |
| Baby 15 | 30 wks 6d | VB | Female | 1.3 | MF | Yes |
| Baby 16 | 32 wks | CSB | Male | 1.9 | MF | Yes |
| Baby 17 | 40 wks 2 d | CSB | Female | 3.6 | EBF | No |
| Baby 18 | 40 wks 2d | VB | Female | 2.54 | MF | No |
| Baby 20 | 38 wks | VB | Male | 3.7 | MF | No |
| Baby 21 | 40 wks | VB | Female | 2.2 | EBF | No |
| Baby 22 | 39 wks | CSB | Male | 3 | EBF | Yes |
| Baby 23 | 33 wks 2 d | VB | Female | 2.25 | EBF | Yes |
| Baby 24 | 40 wks 3 d | VB | Female | 3.1 | EBF | No |
| Baby 25 | 34 wks 3 d | VB | Male | 3.4 | EBF | No |
| Baby 26 | 37 wks 4 d | VB | Female | 3.5 | EBF | No |
| Baby 27 | 37 wks 2 d | CSB | Female | 2.5 | MF | No |
| Baby 28 | 37 wks 2 d | CSB | Male | 2.55 | MF | No |
| Baby 30 | 34 wks | VB | Female | 2.9 | EBF | No |

wks- weeks, d- day, CSB- Caesarean Section Birth, VB- Vaginal Birth, MF- Mixed Fed, EBF-Exclusive Breast Fed

## Sample collection and processing

Faecal samples were obtained monthly from 28 infants for the first year of life with a varying number of samples collected from individual babies (S1 Table), making 169 faecal samples in total. Faecal samples were obtained from the baby's diapers immediately after defecation and transferred into a sterile 20 ml sample bottle. Fifteen ml of absolute ethanol was added into the faecal bolus, and the samples left at room temperature for 24 hours, after which the ethanol was decanted carefully, ensuring the bolus remained intact. The dehydrated faecal bolus was air dried and transferred to another sterile tube containing 15 ml of 3 mm-sized silica gel beads (Guangdong Guanghua Sci-Tech Co. Ltd, Guangzhou city, China). A cotton wool wad was placed on top of the silica gel bead to prevent contact between the silica gel and the faecal bolus. This method has been described previously for sample collection in similar settings [20–22]. The samples were then stored at room temperature until shipment to the Rowett Institute, University of Aberdeen, Aberdeen, Scotland where they were kept at 4˚C until further analyses.

## Total DNA extraction and the 16S rRNA gene library preparation

Total DNA was extracted from 0.1g (dry weight) of each faecal sample using the FastDNA™ SPIN kit (MP Biomedicals, USA) following the manufacturer's instructions as previously described by [23]. The V1-V2 region of the 16S rRNA gene was amplified using fusion bar-coded primers as previously described [23] 27f_Miseq 5′–**AATGATACGGCGACCACCGA GATCTACAC***TATGGTAATTCC*AGMGTTYGATYMTGGCTCAG–3′ and 338R_MiSeq 5′– **CAAGCAGAAGACGGCATACGAGAT** nnnnnnnnnnnn*AGTCAGTCAGAA*GCTGCCTCCCG TAGGAGT–3′, where the bold type indicates the adaptor sequences, italicized are the linkers and the (n) string corresponds to the sample-specific molecular identifier barcodes. The primer sequence annealing to the 16S rRNA gene is in plain text. The PCR amplification was carried out using Q5 High-Fidelity PCR kit (New England Biolabs, Ipswich, Massachusetts, USA). The 25 μl reaction conditions was set at one cycle of 98˚C for two mins, followed by 20 cycles of 98˚C for 30 s, 50˚C for 30 s, 72˚C for 90 s, 72˚C for five mins and held at 10˚C. Each PCR was done in quadruplicate to obtain sufficient material for subsequent sequencing. Following amplification, the replicate products were pooled together and a 10 μl aliquot with 2 μl of 6X loading dye (New England Biolabs, Ipswich, Massachusetts, USA) was checked on 1% agarose gel. Amplicons were cleaned up by ethanol precipitation through the addition of 0.3 volumes of 1M NaCl and 2 volumes of 100% Ethanol, stored overnight at -20˚C, and centrifuged at 4˚C at 14000-16000g for 20 mins. Pellets were washed with 600 μl of cold 70% ethanol, re-pelleted, air-dried and resuspended in 30 μl of TE buffer and kept at 4˚C overnight. Quantification of DNA samples was performed using Qubit 2.0 Fluorometer (Life Technologies). All samples were pooled into a final tube containing an equimolar mix of each amplicon, and a final clean up step was performed using AMPure XP magnetic beads (Beckman Coulter). The subsequent sequencing was performed on the Illumina MiSeq platform producing 300 bp paired end sequencing data, generating between 3251 and 41103 reads per sample (average 15405). These sequences were deposited in the European Nucleotide Archive (ENA) under the Accession number PRJEB31073.

## Bioinformatics analysis of the sequence reads

The quality of the sequences was assessed using FastQC (version 0.11.3) [24] and all were found to be of average quality with no presence of adaptor sequence. The generated raw sequences underwent downstream bioinformatics analysis with DADA2 (version 1.3.1) [25] using default parameters to quantify sequence variants and assign taxonomy based upon the

GreenGenes 13.8 database. Singleton sequence variants (those present in only a single sample at a single count) were removed to leave 1619 sequence variants. The sequence table was converted to Biom format and sequence variant abundances were summarised using Biomformat (version 2.1.3) [26].

Diversity analysis was performed using the core_diversity_analyses.py script from QIIME (version 1.9.0) [27] with a subsampling level of 1993, which enabled all samples to be kept for analysis. Rarefaction curves plateau at this subsampling depth, demonstrating that this was sufficient to capture the sample diversity (S1 Fig). Core diversity analyses calculated five alpha diversity measures (observed species, Chao [28], Shannon Index [29], and Simpson Index [30] and two beta diversity measures Bray Curtis [31] and Binary Jaccard [32].

Statistical testing of stratification of samples by meta data category was performed using the adonis statistical test [33] on the Bray Curtis diversity metrics, implemented by the compare categories script from QIIME (version 1.9.0) [27]. Differential abundance testing of amplicon sequence variants (ASVs) between groups was carried out by converting the biom file to a PhyloSeq object [34] and testing differential abundance using DESeq2 (version 1.14.1) [35]. LEfSe analysis [36] was performed to confirm DeSeq2 results. This analysis was done using the Huttenhower Galaxy server (http://huttenhower.sph.harvard.edu/galaxy/). Only taxa that had Linear Discriminate Analysis score (LDA > 2) were considered as significantly different between the groups.

## Detecti on of *Bifidobacterium* in formula milk by genus-specific PCR

The most common formula milk used for mixed feeding was NAN 1, which the manufacturer claimed was fortified with a *Bifidobacterium* strain, "Bifid LAL". In order to verify this by PCR, DNA was extracted from 0.6 g of NAN 1 formula milk powder using the FastDNA™ SPIN kit (MP Biomedicals, USA) following the manufacturer's instructions. The generic *Bifidobacterium* primer set, Bif164f (GGGGTGGTAATGCCGGATG) and Bif662r (CCACCGTTACACCGGGAA) were used for the amplification [37]. The PCR machine was programmed as follows; initial denaturation of the DNA template at 94˚C for 5 mins, 35 further cycles of denaturation at 94˚C for 30 sec, annealing at 62˚C for 20 sec and extension at 68˚C for 40 sec, with final extension of 68˚C for 7 mins. The size of the product was estimated on 1.2% agarose gel in Tris-borate EDTA buffer (TBE) at 120V for 30 mins and visualized under the UV light.

## Identification of short chain fatty acids (SCFAs) in faecal samples

Short chain fatty acids (SCFAs) present in each infant faecal sample were identified by gas chromatography. Weighed dried samples (150 mg) were resuspended in sterile distilled water (1.1mL) and the concentration of SCFAs measured in duplicate in 1:16 dilution of 250 μl supernatant. Samples were prepared as described previously [38]. Briefly, 25 μl of internal standard, 0.25 ml HCl and 1 ml of ether was added to 0.5 ml of diluted sample. Standards were included in each analysis set and contained 25 μl of internal standard, 0.5 ml of external standard, 0.25 ml HCl and 1 ml of ether. All tubes were vortexed for 1 min and centrifuged for 10 min at 3000 rpm. The upper ether layer was then transferred to a new tube and the sample was re-extracted with a further 0.5 ml of ether. Samples were vortexed and centrifuged as before and the ether layers combined. 400 μl of the combined ether layer was transferred to a Wheaton screw cap vial containing 50μl of MTBSFA (Sigma Aldrich). Samples were heated at 80˚C for 20 min, then left at room temperature for 48 hr to derivatize before being run on a Hewlett Packard (Palo Alto, CA, USA) gas chromatograph (GC) fitted with a fused silica capillary column, with helium as the carrier gas.

## Results

### Overall taxonomic profiles of the infant gut microbiota

The protocol used to collect the samples relied on desiccation to preserve the samples prior to shipment and further processing. We validated the method comparing the bacterial composition profiles of samples from two volunteers processed fresh, or stored frozen at -70˚C or desiccated for 3 months before DNA extraction. Although a few differences were observed in taxa detected, these were primarily driven by inter-individual variation rather than processing method (S2 Fig). This supported the previous validation data when bacterial and SCFA abundance profiles of replicate samples stored either desiccated or frozen at -80˚C [21] were compared.

A total of 169 faecal samples were analysed with an average of seven samples per infant. Biometric information about the babies at birth; the sex, delivery mode, feeding method, gestational age, birth weight, antibiotic treatment (if given) are provided in Table 1. Microbial composition analysis shows that the phylum Actinobacteria was the most abundant across all time points (45.95 ± 2.12; Mean ± SEM), with lower abundance of Firmicutes (36.73 ± 1.78), Proteobacteria (12.96 ± 1.47), Bacteroidetes (4.30 ± 0.57) and Fusobacteria (0.05 ± 0.03) (S2 Table). The remaining phyla (Tenericutes, Verrucomicrobia, Cyanobacteria and Lentisphaerae), comprised less than 0.01% of the total microbiota, and are grouped together as Others. Taxonomic profiles of samples grouped by time point at the level of Phylum showed a general decrease in Proteobacteria by the third sample at month 2 (t2) and a slight increase in Bacteroidetes as time points progressed (Fig 1).

At the assigned genus level, across all samples and time points, *Bifidobacterium* species were the most abundant representing 41.25% of the total genera identified. This was followed

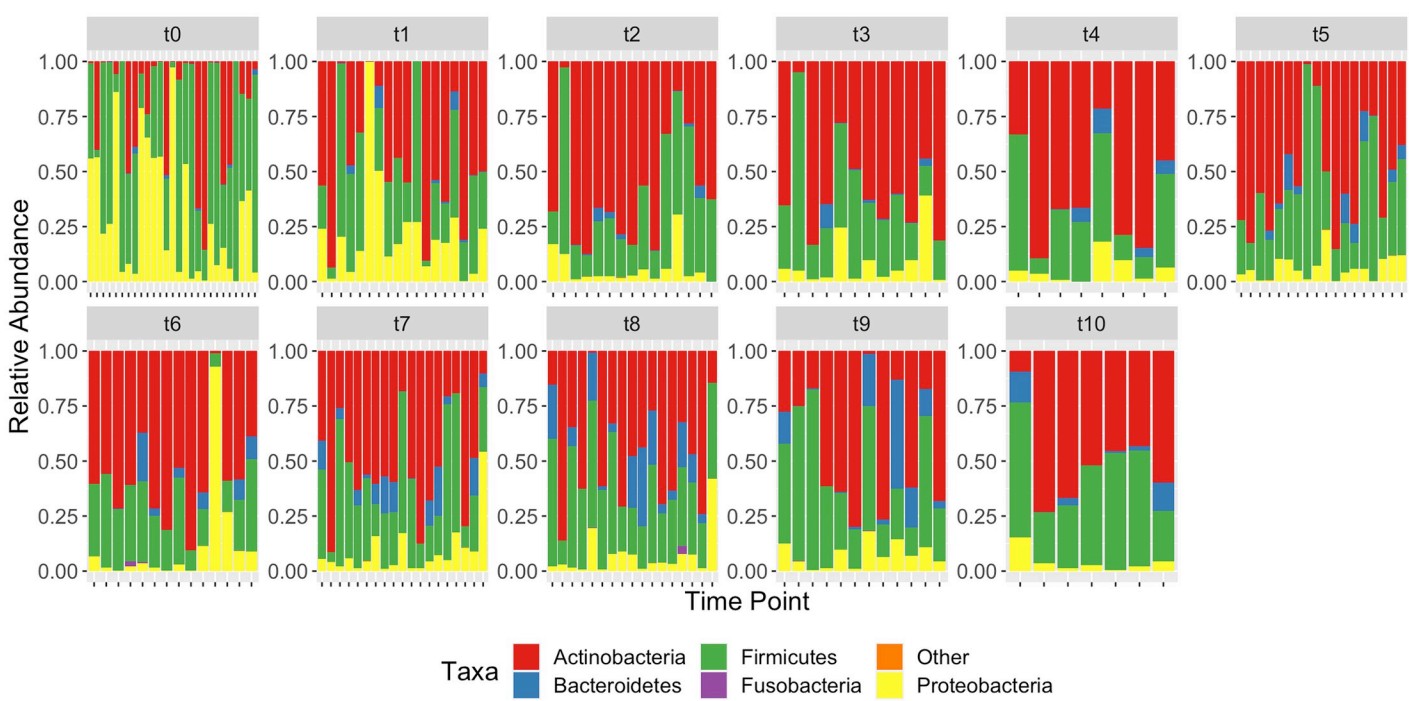

**Fig 1. Taxonomic profile showing the infants' gut microbial composition at phylum level grouped by time points.** Samples at birth (or within one week of life) are represented by t0 and months 1–10 are represented by t1, t2, t3, t4, t5, t6, t7, t8, t9, and t10 respectively. The major phyla shown are Actinobacteria (red), Proteobacteria (yellow), Firmicutes (green), Bacteroidetes (blue), Fusobacteria (purple) and others (orange).

by *Streptococcus* 15.86%, *Trabulsiella* 5.46%, *Klebsiella* 4.64%, *Bacteroides* 3.49%, and *Enterococcus* 3.24% species. All other genera were below 3% of the total sequence reads (S3 Table). Interestingly only 28 of the total 169 samples contained <5% *Bifidobacterium* species, and 19 of these were the initial samples taken within 1 week of birth.

## Microbial differences between the natural birth and caesarean section birth infants

There was no statistically significant difference in the alpha diversity or taxonomic profile between the birthing methods when all samples were grouped together across time points. However, differences were observed in the relative abundance of some taxa when all samples collected within 7 days of birth were compared. In total 21 different sequence variants were significantly different following DESeq2 analysis, with four higher in VB babies, although these were all single sequence variants of that taxa. Seven of the single sequence variants more abundant in CSB babies were identified as *Clostridium perfringens*, three as *Trabulsiella farmeri* and two in the *Klebsiella* genus giving confidence that these taxa, all classed as enteric pathogens, genuinely had higher abundance following CSB (S4 Table). The only sequence variant classified as *Bifidobacterium animalis* was also significantly increased in CSB samples (S4 Table). This identification of *B. animalis* in four CSB babies and no VB babies in the first seven days was further supported by LEfSe analysis. Although one sequence variant identified as *Bifidobacterium longum* was 24-fold more abundant in VB compared to CSB babies, other ASVs also characterised as *B. longum* were unchanged.

## Microbial differences related to infant's age and diet

The diversity of the microbiota increased with age. The alpha diversity showed that there was a rise in the sample richness across timepoints as infant age increased (Fig 2). This increased diversity within samples over time could be linked to diet. Nigerian babies are weaned with the introduction of the local diet usually referred to as "*Ogi*". This initial weaning food is a fermented semi-solid slurry made from maize, millet and sorghum, and thereafter solid foods such as rice, yam flour made into bolus with "*Ewedu*" soup (Jute leaf), bean and bean products are introduced. Microbial diversity also significantly increased after weaning (the local/solid fed samples LF/SF) across all time points (p < 0.01; Kruskal Wallis followed by Dunn Post-Hoc analysis; Fig 3). Preweaning samples from both exclusive breast-fed (EBF) and mixed fed (MF) babies had similar diversity.

The beta diversity revealed that the differences between individuals also increased with age. The samples were grouped by time point for clearer visualisation of age-related changes: early (t0-t4), mid (t5-t8) and late (t9-12). Most of the early time points form a tight cluster, illustrating considerable relatedness (p = 0.001) in the taxa present in these pre-weaning time points, irrespective of EBF or MF status. The later timepoints are much more scattered, demonstrating the increased inter-individual diversity (Fig 4).

Investigation into the differences between specific taxa between preweaning EBF and MF samples using DESeq2 revealed that 23 sequence variants had a significant difference in abundance (adjusted p value < 0.05; Table 2). This comparison focused on samples from timepoints t1-t3, removing any samples below 1 month of age (t0) where the microbiota is still establishing, and any samples after weaning had already commenced. A significant increase in sequence variants classified as *Bifidobacterium bifidum* and *Bifidobacterium breve* were identified in mixed fed samples (shown by the negative fold change). Additional LEfSe analysis supported these observations. Three ASVs identified as *Blautia* species were also increased in abundance in MF compared to EBF babies (Table 2). In contrast, in exclusive breast-fed

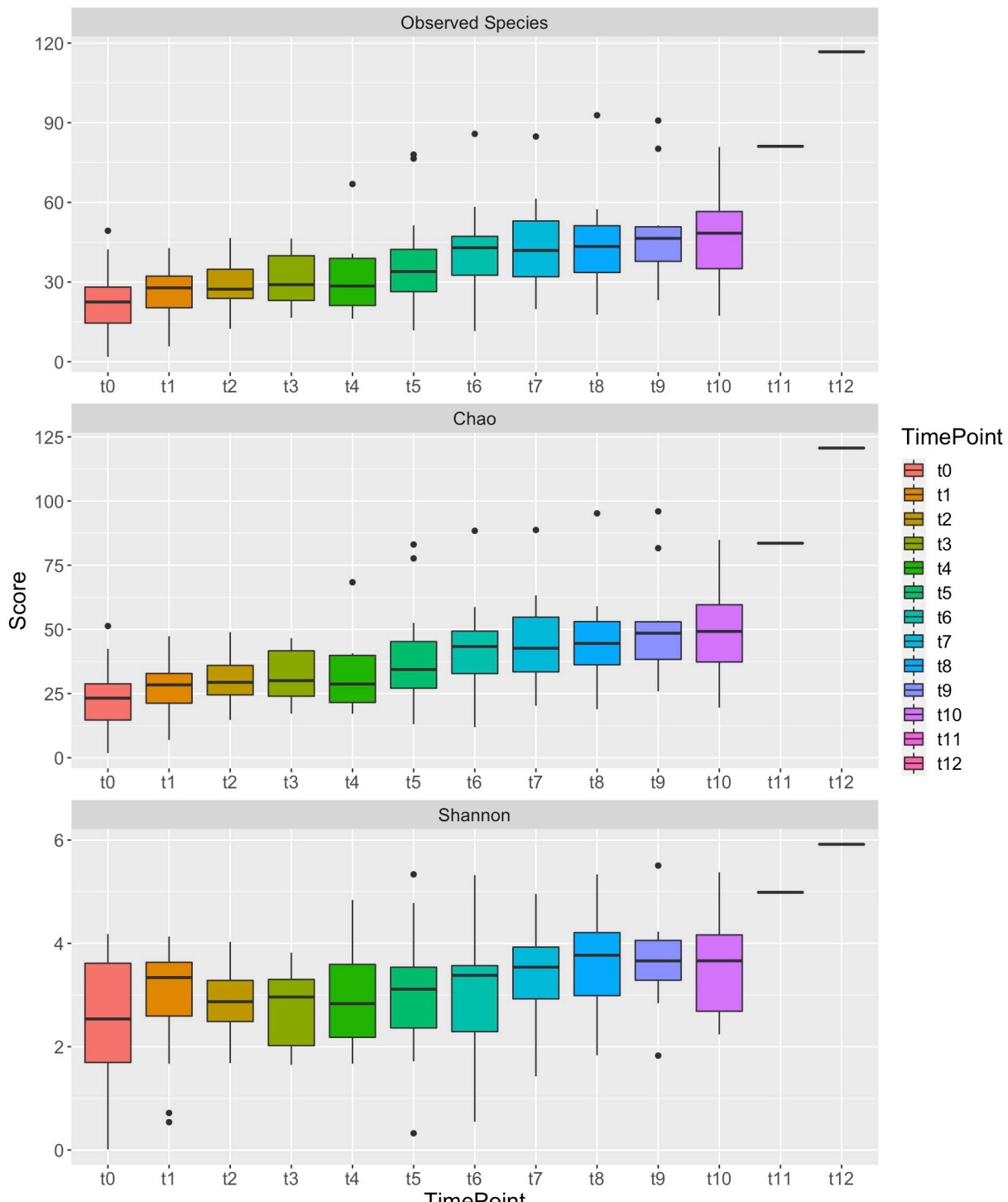

**Fig 2. Change in alpha diversity over time of samples collected from all babies.** Samples grouped by time points showing the increasing diversity of the gut microbial composition with age. The preweaning samples were significantly different to the samples collected after weaning commenced (LF/SF) for each diversity metric shown (Observed Species p = $1.412^{-07}$, Chao p = $9.013^{-08}$, and Shannon p = 0.01055; Kruskal Wallis analysis). Samples from a single baby were available for timepoints t11 and t12.

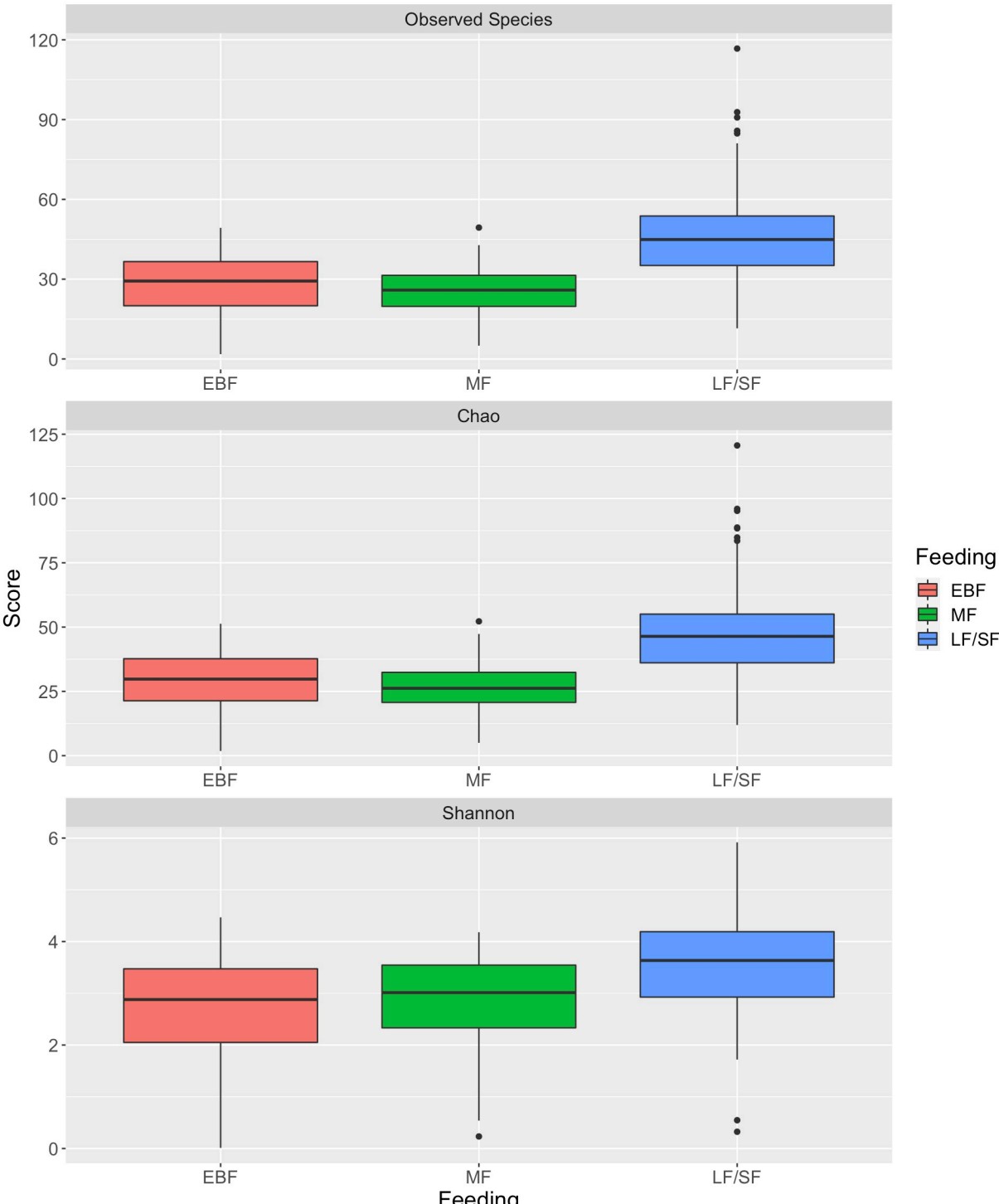

**Fig 3. Alpha diversity linked to feeding method for samples collected from all babies.** Samples were grouped into EBF (exclusive breastfeeding), MF (mixed feeding: breast milk plus formula milk) and weaning samples following the introduction of local food plus solid foods (LF/SF). EBF and MF samples were significantly different to the LF/SF samples (p < 0.01; Kruskal Wallis followed by Dunn Post-Hoc analysis).

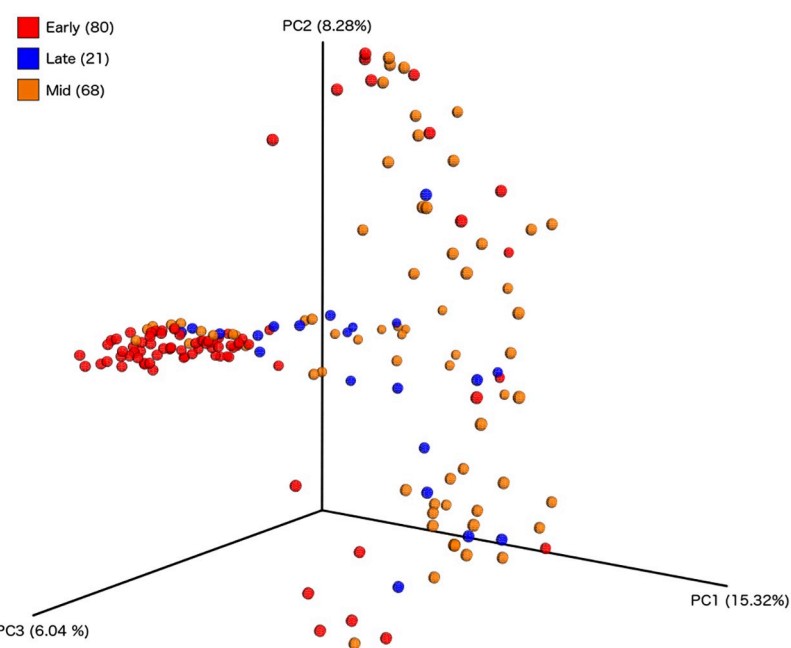

**Fig 4. Principal Component Analysis showing similarity between samples from different babies at grouped timepoints.** Principal Component Analysis plots based upon Bray Curtis Diversity indices for babies at grouped timepoints. Early time points (t0-t4, n = 80) are shown in red, middle time points (t5-t8, n = 68) in orange and late time points (t9-10, n = 21) in blue. Statistical testing of sample stratification (adonis test) showed significant clustering by age both when broadly grouped, and when more finely grouped by months (p = 0.001 for both tests).

samples, the *Bifidobacterium* species with increased abundance (positive fold change) were *Bifidobacterium adolescentis* and *Bifidobacterium longum*. Further investigation confirmed that the latter ASV could be identified as *B. longum* subsp. *infantis*. Sequence variants identified as *Ruminococcus gnavus* and from the genera *Collinsella* and *Sutterella* were also more abundant in EBF babies.

The most common formula milk used for the 15 MF babies was NAN 1. Interestingly, *Bifidobacterium* was detected in the formula milk (S3 Fig), supporting the manufacturer's claims of fortification, which may partly influence the higher abundance of the genus in the mixed fed samples. *Bifidobacterium* was not detected in the un-supplemented powdered milk routinely consumed by adults in Nigeria.

Comparisons were also made between the abundance of sequence variants in the grouped preweaning samples (excluding those from <1 month of age), regardless of whether they were from EBF or MF babies, versus all samples after the introduction of local and solid food (the LF/SF group). This identified 22 significantly different sequence variants (adjusted p value < 0.05; Table 3), with just five increased before weaning including three *Streptococcus* ASVs and one ASV from the genus *Propionibacterium*. Sequence variants with a significantly higher abundance after the introduction of solid food included several identified as *Bifidobacterium* spp., *Bacteroides fragilis*, *Megamonas* spp., *Coprococcus* spp., and variants from the Clostridiaceae family (adjusted p value < 0.05) (Table 3). A single *Lactobacillus mucosae* variant was also more abundant following weaning. Overall, *Lactobacillus* ASVs were identified in 50% of samples, representing an average of 1.93% of identified ASVs (S3 Table). More than 70% of these samples were from timepoint 4 onwards, and only 18% of samples lacking *Lactobacillus* were from timepoint 8 or later (with 9/15 of these samples from three infants where *Lactobacillus* was never detected).

**Table 2. Sequence variants with significant differential abundance between exclusive breastfed (EBF) and mixed fed (MF) samples (DESeq2; adjusted p < 0.05).**

| Sequence Variant ID | LogFC* | Adjusted P Value | Phylum | Family | Genus | Species |
|---|---|---|---|---|---|---|
| seq10 | 28.43 | 1.11E-20 | Firmicutes | Streptococcaceae | *Streptococcus* | |
| seq33 | 25.14 | 1.40E-16 | Actinobacteria | Bifidobacteriaceae | *Bifidobacterium* | *adolescentis* |
| seq42 | 25.09 | 6.56E-35 | Firmicutes | Lachnospiraceae | *[Ruminococcus]* | *gnavus* |
| seq35 | 24.98 | 2.84E-17 | Firmicutes | Lachnospiraceae | *[Ruminococcus]* | *gnavus* |
| seq97 | 24.67 | 2.21E-21 | Firmicutes | Streptococcaceae | *Streptococcus* | |
| seq63 | 24.14 | 2.29E-20 | Proteobacteria | Alcaligenaceae | *Sutterella* | |
| seq52 | 23.78 | 6.10E-22 | Proteobacteria | Enterobacteriaceae | *Trabulsiella* | |
| seq32 | 23.47 | 1.36E-14 | Actinobacteria | Coriobacteriaceae | *Collinsella* | *aerofaciens* |
| seq87 | 23.34 | 1.83E-14 | Actinobacteria | Bifidobacteriaceae | *Bifidobacterium* | *adolescentis* |
| seq39 | 22.94 | 3.94E-14 | Firmicutes | *Erysipelotrichaceae* | | |
| seq170 | 6.62 | 0.00637 | Firmicutes | Staphylococcaceae | *Staphylococcus* | *aureus* |
| seq2 | 5.7 | 0.02054 | Actinobacteria | Bifidobacteriaceae | *Bifidobacterium* | *longum* |
| seq22 | -7.41 | 0.04425 | Actinobacteria | Bifidobacteriaceae | *Bifidobacterium* | *breve* |
| seq50 | -8.42 | 0.00236 | Firmicutes | Lachnospiraceae | *Blautia* | |
| seq58 | -22.49 | 1.25E-13 | Proteobacteria | Enterobacteriaceae | *Trabulsiella* | *farmeri* |
| seq54 | -23.26 | 2.14E-16 | Proteobacteria | Enterobacteriaceae | *Klebsiella* | |
| seq194 | -23.73 | 1.63E-19 | Firmicutes | Streptococcaceae | *Streptococcus* | |
| seq96 | -24.32 | 1.70E-20 | Firmicutes | Lachnospiraceae | *Blautia* | |
| seq84 | -24.52 | 1.13E-16 | Firmicutes | Lachnospiraceae | *Blautia* | |
| seq31 | -26.13 | 7.18E-18 | Actinobacteria | Bifidobacteriaceae | *Bifidobacterium* | *bifidum* |
| seq64 | -26.42 | 2.26E-22 | Firmicutes | Veillonellaceae | *Veillonella* | |
| seq18 | -26.52 | 1.73E-28 | Actinobacteria | Bifidobacteriaceae | *Bifidobacterium* | *bifidum* |
| seq26 | -27.95 | 3.82E-23 | Actinobacteria | Bifidobacteriaceae | *Bifidobacterium* | |

* Positive log fold change (logFC) indicates higher expression in EBF samples, Negative log fold change indicates higher expression in MF samples (adjusted p value < 0.05).

Taxonomy classification using DESeq2: phylum, family, genus, species.

Where category is blank the ASV could not be differentiated to that taxonomic level.

### Concentrations of faecal short chain fatty acids change with age and diet

The detection of short chain fatty acids in the infant faecal samples reflects the microbial and dietary changes over age. Acetate and butyrate proportions increased significantly following weaning, presumably reflecting the incorporation of more fermentable carbohydrates into the diet (p<0.0001, Fig 5). In contrast proportions of succinate were highest in the first month of age and the proportion of propionate also declined after weaning. Concentrations of the microbial metabolite lactate were high before weaning and remained high through to the ten month timepoint in some babies. There was much more variation in the detection of lactate across all the samples, with 45% of samples containing no detectable lactate.

## Discussion

In this longitudinal study, we investigated the influence of birth method, age, and feeding pattern on the development of the gut microbiota of Nigerian infants. This is the first study to characterise the gut microbial composition of Nigerian infants in such detail. Overall, the infant gut is dominated by the phylum Actinobacteria, specifically the genus *Bifidobacterium*. This is consistent with many previous studies including Canadian infants [39], European infants [17] and Scottish infants [23].

**Table 3. Sequence variants with significant differential abundance between samples collected before and after weaning.**

| Sequence Variant ID | logFC* | Adjusted p value | Phylum | Family | Genus | Species |
|---|---|---|---|---|---|---|
| seq10 | 26.27 | 2.67E-18 | Firmicutes | Streptococcaceae | *Streptococcus* | |
| seq55 | 23.91 | 1.12E-47 | Firmicutes | | | |
| seq27 | 8.05 | 6.41E-05 | Actinobacteria | Propionibacteriaceae | *Propionibacterium* | |
| seq17 | 6.08 | 0.005516 | Firmicutes | Streptococcaceae | *Streptococcus* | *luteciae* |
| seq79 | 5.72 | 0.008479 | Firmicutes | Streptococcaceae | *Streptococcus* | |
| seq120 | -5.07 | 0.047676 | Firmicutes | Streptococcaceae | *Streptococcus* | |
| seq109 | -5.12 | 0.020535 | Firmicutes | Lachnospiraceae | *Coprococcus* | |
| seq14 | -5.23 | 0.000774 | Actinobacteria | Bifidobacteriaceae | *Bifidobacterium* | |
| seq226 | -5.62 | 0.049114 | Firmicutes | Veillonellaceae | *Megasphaera* | |
| seq132 | -5.86 | 0.002481 | Firmicutes | Clostridiaceae | *SMB53* | |
| seq77 | -5.99 | 6.86E-05 | Firmicutes | Clostridiaceae | *02d06* | |
| seq148 | -6.14 | 0.001364 | Firmicutes | Erysipelotrichaceae | *[Eubacterium]* | *dolichum* |
| seq114 | -6.6 | 5.82E-06 | Firmicutes | Clostridiaceae | *SMB53* | |
| seq73 | -7.19 | 1.15E-05 | Proteobacteria | Pasteurellaceae | *Haemophilus* | *Parainfluenzae* |
| seq16 | -9.6 | 3.46E-09 | Actinobacteria | Bifidobacteriaceae | *Bifidobacterium* | *bifidum* |
| seq51 | -24.4 | 1.76E-55 | Firmicutes | Veillonellaceae | *Veillonella* | *dispar* |
| seq101 | -25.03 | 4.35E-24 | Firmicutes | Lactobacillaceae | *Lactobacillus* | *mucosae* |
| seq61 | -25.28 | 6.74E-35 | Firmicutes | Veillonellaceae | *Megamonas* | |
| seq45 | -25.96 | 2.02E-66 | Firmicutes | Clostridiaceae | *02d06* | |
| seq29 | -27.13 | 1.76E-55 | Bacteroidetes | Bacteroidaceae | *Bacteroides* | *fragilis* |
| seq25 | -28.04 | 4.47E-38 | Actinobacteria | Bifidobacteriaceae | *Bifidobacterium* | |
| seq23 | -28.13 | 3.59E-42 | Bacteroidetes | Bacteroidaceae | *Bacteroides* | *fragilis* |

* Positive log fold change (LogFC) indicates variants more abundant in the pre-weaning group, Negative log fold change indicates variants more abundant in the weaning (LF/SF) group, (adjusted p value < 0.05).

Taxonomy classification using DESeq2: phylum, family, genus, species. Where category is blank the ASV could not be differentiated to that level.

The bacterial diversity increased as the age of the infants increased, particularly following the introduction of solid food during weaning, supporting the hypothesis that the accompanying diet changes have a major impact on the microbial composition and diversity. The early samples (0–4 months) clustered closely together, with more inter-individual variation

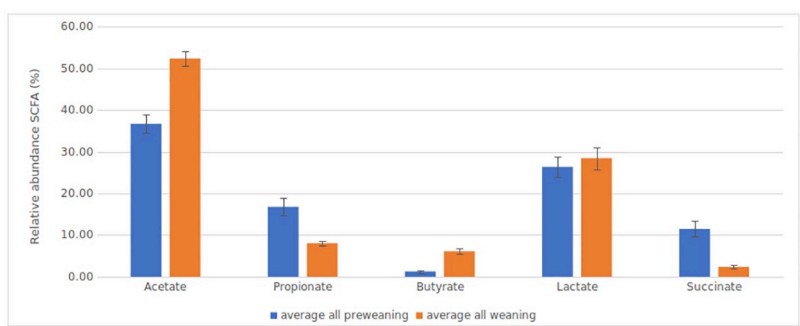

**Fig 5. Relative abundance of SCFAs detected in faecal samples comparing pre-weaning and weaning infants.** The relative abundance of individual SCFAs (as a percentage of the total SCFA concentration), measured in duplicate samples from all babies during pre-weaning (blue) and weaning (orange) are plotted with accompanying error bars (standard error of the mean). Differences were calculated using a Wilcoxon Rank test of means before and after weaning commenced, indicating highly significant increases in acetate and butyrate (p<0.001), and significant decreases in succinate (p<0.05).

apparent in the samples from the older babies. Different factors such as developmental changes (oral stage), change in diet (introduction of solid foods), and familiarity with environment (crawling and toddling) contribute to the higher microbial diversity in the older infants, although the timing of these changes differs between infants. Similar chronological changes in the composition and diversity of the microbiota have been observed in single time point studies focused on different infants at specific ages [13, 14]. Two other longitudinal studies have also shown increased bacterial diversity with increasing age, although these focused on slightly older infants, with timepoints from 9 months (Danish infants; [40]) or 1 year (Irish infants; [41]) onwards. The increased diversity of the maturing microbiota coincides with increased metagenomic functionality [15].

Some differences were observed in the presence of specific taxa between samples obtained within the first week of life from vaginal birth and caesarean section birth babies, with bacteria not usually linked to a healthy gut more prevalent in CSB babies. The impact of birth method on the microbial composition has been shown previously [12, 16, 42–44], although other studies found no difference [45, 46], perhaps reflecting the exact timepoints of the samples compared. Excluding the first samples from our comparisons of the preweaning and postweaning microbiota changed the differentially abundant groups identified, due to the prevalence of Proteobacteria in the very early samples before the microbiota stabilised. Many of the first colonisers in the Nigerian infants were facultative anaerobes. Members of the genera *Streptococcus* and *Enterococcus* were present in the first samples and persisted for three months, while *Staphylococcus*, *Trabulsiella*, and *Klebsiella* disappeared after two months. These genera belong to the phyla Proteobacteria and Firmicutes, and members of these groups have previously been reported as early infant gut colonisers [11, 12, 46]. The impact of delivery mode on the composition of infant gut microbiota has been shown to persist for as much as four years [41].

*Bifidobacterium* was the dominant bacterial genus, comprising more than 40% of the total bacteria identified across all samples at all timepoints, consistent with many other studies. An early study comparing the microbial composition in 6 month old Malawian and Finnish infants (using fluorescent in situ hybridisation and qPCR techniques) also observed a dominance of *Bifidobacterium* in both cohorts [47]. We found that different species of *Bifidobacterium* were prevalent in EBF (*B. adolescentis*, *B. longum* subsp. *infantis*) compared to MF (*B. bifidum*, *B. breve*) infants. This is consistent with the prevalence of *B. longum* subsp. *infantis* in breastfed Finnish, Estonian and Russian infants [15]. It is known that different *Bifidobacterium* species have different specific abilities to utilize human milk oligosaccharides, and that the composition of oligosaccharides is very different between breastmilk and formula milk [48], which may drive this difference. In addition, the NAN 1 formula milk consumed by most (80%) of the MF babies is fortified with a strain of *Bifidobacterium*, although we could not confirm if this was the same strain detected in our MF infants. Interestingly, the increased prevalence of *B. bifidum* persisted through to the samples after weaning. A *B. bifidum* strain identical to a commercial isolate commonly added to baby formulas in Russia was prevalent in samples from Russian babies but not Finnish or Estonian infants analysed in the same study [15]. The abundance of *Collinsella aerofaciens* and *Sutterella*, in the exclusively breastfed (EBF) infants is consistent with other studies of breastfed infants [49, 50]. The increased prevalence of *R. gnavus* in EBF compared to mixed fed infants, preweaning, may reflect the ability of this bacterium to grow on fucosylated glycans such as those found within human milk oligosaccharides as well as host mucins [51].

The differences in the microbial composition before and after weaning commenced is at least partly due to the shift from milk to solid foods. The indigenous Nigerian weaning diet consists of the local fermented food (*Ogi*), which is rich in fibre and vitamins and also contains *Lactobacillus* species that are potent against some pathogenic *E. coli* strains [52]. The weaning

samples were characterized by increased bacterial diversity, as the infant microbiota matures towards an adult-like composition, and emergence or increased abundance of specific bacterial signatures including some with active carbohydrate metabolic activity, specifically members of the families Lachnospiraceae, uncultured Clostridiaceae and Lactobacillaceae. The changes that occurred in the gut microbiota of these Nigerian infants, before and after weaning are consistent with those reported in other studies during the transition from milk-dependent food to solid food [15, 53, 54]. The persistence of *Bifidobacterium* species after weaning differs from studies of older African children where a low abundance of *Bifidobacterium* was observed [14, 22], presumably reflecting the fact that our older infants (<1 yr old) still received milk feeding.

Members of the Bacteroidetes phylum also increased during weaning, and *Bacteroides* species are known to possess many genes encoding carbohydrate-degrading enzymes. Sequence types classed as *B. fragilis* in particular increased in abundance in some babies. This observed increased level of *Bacteroides* species agrees with the findings of [17] who found higher *Bacteroides* in European infants during weaning. Of note from our data was the finding that 25% of samples contained an ASV identified as *Prevotella* and 22/39 of these samples were post-weaning samples. A higher abundance of *Prevotella* compared to *Bacteroides* species is characteristic of African compared to European children samples [55]. A higher abundance of *Prevotella* was identified in Bassa individuals (both adults and children under 3 yr of age) from a rural northern settlement in Nigeria compared with urban dwelling Nigerians [22].

Dietary changes also affect the production of bacterial fermentation products, including short chain fatty acids and lactate. The relative abundance of the fermentation products acetate and butyrate increased during weaning, as would be expected following the introduction of solid foods. These acidic fermentation products help reduce the intestinal pH, helping to protect against the proliferation of pathogenic organisms. Lactate and acetate were dominant at all timepoints, consistent with the prevalence of *Bifidobacterium* species. The persistence of lactate after weaning starts is potentially due to the inclusion of the fermented *Ogi* during weaning, either by the lactate content or the presence of *Lactobacillus* species in the product. It would be interesting to ascertain if the *Lactobacillus* species identified in these infants corresponded to those found in the *Ogi*.

Notable in this study is the significant increase in butyrate when weaning starts between 4–6 months. This is likely to be related to the introduction of solid foods to the infant diet, and the change in microbial composition. Increased short-chain fatty acids concentrations have been reported in African children with distinct production of butyrate, different from their European counterparts [14]. More recently Nilsen et al. linked increased butyrate between infants of 6 and 12 months to increased prevalence of the butyrate producing bacteria *Eubacterium rectale* and *Faecalibacterium prausnitzii* [56], but neither of these species were detected at high abundance in our samples.

In summary, this study has provided baseline information on the gut microbial composition of Nigerian infants. The observed taxonomic differences in the gut microbiota between preweaning and weaning samples in Nigerian infants, as well as butyrate production, were influenced by diet. Introduction of solid foods encouraged an increase in microbial diversity, helpful for a healthy life.

## Supporting information

**S1 Fig. Rarefaction curves showing A) Observed species, B) Chao, and C) Shannon diversity index over subsampling depths.** Analysis was performed using the core diversity analyses.py script from QIIME (version 1.9.0).
(TIF)

**S2 Fig. Bacterial composition profiles of samples from two volunteers to validate the desiccation storage method.**
(DOCX)

**S3 Fig. The presence of *Bifidobacterium* in NAN 1 formula milk.**
(DOCX)

**S1 Table. Timing of faecal samples collection from individual babies (n = 28) during the first year of life.** Samples immediately after weaning commenced are shaded and shown in bold.
(XLSX)

**S2 Table. Percentage abundance of phyla in all samples from all babies, across all time points.**
(XLSX)

**S3 Table. Percentage of most abundant (top 39) genera in all samples across all time points, in all samples from all babies.**
(XLSX)

**S4 Table. Differentially abundant ASVs following DESeq2 analysis, comparing samples at day 7 between vaginal birth (+ve change) and C-section birth (-ve change) babies.** Negative fold changes indicate ASVs (17) more abundant in the C-section birth babies, and positive fold changes indicate the four ASCs more abundant in vaginal birth babies. ASVs are identified to the genus or (when possible) species level.
(XLSX)

## Acknowledgments

We would like to thank our recruited Mums and babies for providing the samples without which this study would not have been possible. We thank the Centre for Genome Enabled Biology and Medicine for Illumina sequencing and useful discussions.

## Author Contributions

**Conceptualization:** Omolanke T. Oyedemi, Funmilola A. Ayeni, Karen P. Scott.

**Data curation:** Omolanke T. Oyedemi, Sophie Shaw.

**Formal analysis:** Sophie Shaw, Karen P. Scott.

**Funding acquisition:** Funmilola A. Ayeni, Karen P. Scott.

**Investigation:** Omolanke T. Oyedemi, Jennifer C. Martin.

**Methodology:** Omolanke T. Oyedemi, Jennifer C. Martin.

**Project administration:** Funmilola A. Ayeni, Karen P. Scott.

**Software:** Sophie Shaw.

**Supervision:** Funmilola A. Ayeni.

**Validation:** Omolanke T. Oyedemi, Funmilola A. Ayeni, Karen P. Scott.

**Visualization:** Omolanke T. Oyedemi, Sophie Shaw, Jennifer C. Martin, Karen P. Scott.

**Writing – original draft:** Omolanke T. Oyedemi, Sophie Shaw, Karen P. Scott.

**Writing – review & editing:** Omolanke T. Oyedemi, Sophie Shaw, Jennifer C. Martin, Funmilola A. Ayeni, Karen P. Scott.

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
