## [Decision Letter · Decision Letter 0]

23 Nov 2021

PONE-D-21-33595Changes in the gut microbiota of Nigerian infants within the first year of lifePLOS ONE

Dear Dr. Scott,

Thank you for submitting your manuscript to PLOS ONE. After careful consideration, we feel that it has merit but does not fully meet PLOS ONE’s publication criteria as it currently stands. Therefore, we invite you to submit a revised version of the manuscript that addresses the points raised during the review process.

Your manuscript has been evaluated by two reviewers. Both found the manuscript and topic interesting. However, they identified some issues with the current version as you can see in their comments. Both of them actually suggest a more in depth comparative analysis with observations described in literature. I recommend you to submit a revision in which all suggestions by the reviewers have been addressed.

We look forward to receiving your revised manuscript.

Kind regards,

Erwin G Zoetendal, PhD

Academic Editor

PLOS ONE

Journal Requirements:

Reviewers' comments:

Reviewer's Responses to Questions

**Comments to the Author**

1. Is the manuscript technically sound, and do the data support the conclusions?

Reviewer #1: Partly

Reviewer #2: Partly

2. Has the statistical analysis been performed appropriately and rigorously? 

Reviewer #1: No

Reviewer #2: Yes

3. Have the authors made all data underlying the findings in their manuscript fully available?

Reviewer #1: Yes

Reviewer #2: Yes

4. Is the manuscript presented in an intelligible fashion and written in standard English?

Reviewer #1: Yes

Reviewer #2: Yes

5. Review Comments to the Author

Reviewer #1: This manuscript reports the results of a longitudinal gut microbiome study in Nigerian infants in the first months of life. While the uniqueness of the data is clear, the manuscript is really a "by-the-numbers" descriptive gut microbiome paper. I think the authors brush aside a wealth of similar datasets that have indeed been reported, but mostly from Western populations. Searching "longitudinal gut microbiome infants" in Pubmed provides hundreds of hits. I advise the authors to look for the most appropriate datasets available (a western study with similar temporal points, some of the few studies which have included infants gut microbiome like the "old" Yatsunenko and Dominguez-Bello papers...) and use these datasets to contextualize their own datasets. Such a contextualization is needed for a better discussion, and I am sure this will provide more exciting leads.

A few more comments below

More details are needed about the statistical analyses. On some figures, nothing is indicated, and it is not clear if this means no statistical difference? Using t-test requires the data to be normal, so this check needs to be stated somewhere...

L105-108 Is the protocol of storing fresh stool in ethanol and at room temperature validated? It seems that such a treatment could alter the gut microbiota rather than preserve it.

L130-131 It is common to use normalization/cleaning kits before pooling amplicons. Ethanol precipitation is not regarded as a really efficient clean-up method, at least not as good than isopropanol, which is as easy to perform...

L190-194 Basic sequence statistics (average and std error per sample for example) are needed to demonstrate that sufficient sequencing depth was reached

Reviewer #2: The manuscript ” Changes in the gut microbiota of Nigerian infants within the first year of life” PONE-D-21-33595 by Oyedemi et al. describes microbiota development in the Nigerian population and contributes to filling gaps in our knowledge in the microbiota development trajectories in non-western population. The contribution is in this respect very valuable. The manuscript is in general well written and reported. However, I have only some concerns about the methodologies and some suggestions for the improvement of manuscript, as follows:

Introduction:

Although there no studies on the development of the infant gut microbial composition over time from African countries, there are some cross-sectional studies investigating microbiota in African children. Please, describe the main findings from these studies and how microbiota was found to differ in African vs other populations -in this respect direct comparisons may not be available, but major differences may be obvious based on the literature.

M&M

Sample collection was done by immersing the sample in ethanol for 24 h and then drying. Are there previous studied comparing microbiota results obtained by using this method vs analysing fresh (or immediately -80C frozen) samples. If yes, please provide the reference. If not, and you would have such results, please provide them in supplementary material. The collection method is very practical and useful especially in low income or field settings and it would be good to know to what extent it has been validated. Is the protocol the same that was used in refs 17 and 18 or modified?

L115: “0.1g of each faecal sample” -add “dry weight” after 0.1 g

L119: The subtitle “Verification of manufacturer’s claim on NAN 1 formula milk” seems like an advertisement and doesn´t well describe the contents of the paragraph. Change to “Detection of bifidobacteria in formula milk by genus-specific PCR”

Has the method for measuring short chain fatty acids (SCFAs) been validated for ethanol treated and dried faecal samples? Ref 34, which is given as reference, investigated fresh faecal samples presumably. Concerning SCFA results, it is critical to know the validation as the analysis can be expected to be affected largely by sample preparation. Without validation the results may not be reliable and thereof should not be presented.

Results:

L222 and L225: The word “upregulated” may not be the best to describe “had higher abundance”. Revise throughout the manuscript.

Discussion:

L338:What about in comparison to previous findings in African children or other non-western populations?

Include in the discussion a general overview on the microbiota development in Nigerian infants as compared to other populations, what is similar and what is different in the general trajectories.

L365: Correct “species of Bifidobacteria” to “species of Bifidobacterium” and genus name in italic font

L367: Correct “Bifidobacterial” to “Bifidobacterium” and genus name in italic font

L392: “Bacteroides” in italic font. Also Prevotella in L.393 and elsewhere. Check throughout the manuscript that genus names are in italic font.

6. PLOS authors have the option to publish the peer review history of their article (what does this mean?). If published, this will include your full peer review and any attached files.

Reviewer #1: No

Reviewer #2: No

---

## [Author Response · Author response to Decision Letter 0]

10 Jan 2022

Dear Professor Zoetendal,

Thank you for providing reviewers comments and editorial consideration of our manuscript: 

“Changes in the gut microbiota of Nigerian infants within the first year of life” for publication, previously submitted to PLOS ONE with reference number PONE-D-21-33595. We appreciate the comments provided and would be grateful if you could consider the revised attached manuscript for publication. In addressing all the comments, including the more in depth comparative analysis suggested, we feel the manuscript has been clarified and improved.

The detailed rebuttal letter follows below and we have attached a marked up and a clean copy of the revised manuscript as instructed.

Furthermore we have addressed the additional requirements for style and file naming, and have checked the main Figures using the PACE tool. 

We have clarified the information about the funding for the study. The appropriate wording for the funding information paragraph is:

OTO was a self-funded PhD student performing her PhD research at the University of Ibadan, Nigeria. OTO received no specific funding for her PhD work except that her 5 month placement at The Rowett Institute was supported by the Scottish Funding Council - Global Challenge Research Fund (Internal Grant reference SF10180). The Rowett Institute (University of Aberdeen) receives financial support from the Scottish Government Rural and Environmental Sciences and Analytical Services (RESAS).

We thank you for considering this revised manuscript for publication, and look forward to hearing from you in due course.

Yours faithfully,

Karen Scott (corresponding author, on behalf of all authors).

Detailed responses to Reviewer Comments to the Author

Reviewer #1: This manuscript reports the results of a longitudinal gut microbiome study in Nigerian infants in the first months of life. While the uniqueness of the data is clear, the manuscript is really a "by-the-numbers" descriptive gut microbiome paper. I think the authors brush aside a wealth of similar datasets that have indeed been reported, but mostly from Western populations. Searching "longitudinal gut microbiome infants" in Pubmed provides hundreds of hits. I advise the authors to look for the most appropriate datasets available (a western study with similar temporal points, some of the few studies which have included infants gut microbiome like the "old" Yatsunenko and Dominguez-Bello papers...) and use these datasets to contextualize their own datasets. Such a contextualization is needed for a better discussion, and I am sure this will provide more exciting leads.

We appreciate the reviewer pointing that we had not considered our data in comparison to existing literature widely enough. The reviewer is correct that there are many such datasets available from western populations so we have identified those most relevant and compared our own data more substantially with them. We have added appropriate additional text to the Introduction and Discussion to put our study in greater context.

A few more comments below

More details are needed about the statistical analyses. On some figures, nothing is indicated, and it is not clear if this means no statistical difference? Using t-test requires the data to be normal, so this check needs to be stated somewhere...

We have added additional data on the statistical analysis to the figures as requested

Figure 1 – this is observational and does not require independent statistical analysis. The relevant statistical data for the basic bioinformatic analysis has been added to the Materials and Methods section. (see response to changes addressing comment referring to L190-194).

Figure 2 –We had previously quoted the general p values for these comparison in the text, but we have now added the specific numbers for each comparison to the Figure legend.

Figure 3 –Statistical testing of the Beta diversity analysis was previously included in the Materials and Methods section. We have added a further sentence clarifying this (and showing the p value) to the figure legend.

Figure 4 –[replacement figure] The data on the SCFA% abundance were compared using the Wilcoxon Rank test (since the data were not normally distributed)and the statistically different abundances are given in the legend 

L105-108 Is the protocol of storing fresh stool in ethanol and at room temperature validated? It seems that such a treatment could alter the gut microbiota rather than preserve it.

This protocol had been previously validated (indicated in the reference cited), and we have clarified this in the text, but was also validated by ourselves. We have added further information about our validation in the supplemental material, as detailed in the response to a similar request from reviewer 2 (M&M comment).

L130-131 It is common to use normalization/cleaning kits before pooling amplicons. Ethanol precipitation is not regarded as a really efficient clean-up method, at least not as good than isopropanol, which is as easy to perform.

The clean-up method that was used prior to sequencing has been clarified in the text. In our method recommended by the University of Aberdeen central sequencing facility (which has been published many times by various groups within the Gut Microbiology lab) the initial cleanup of individual samples relies on precipitation of the DNA amplicons with NaCl and ethanol, with a subsequent cleanup utilising Amp-Pure magnetic beads applied after amplicon pooling into an equimolar pool, prior to performing the sequencing run. This balances cleanup efficacy and expense.

L190-194 Basic sequence statistics (average and std error per sample for example) are needed to demonstrate that sufficient sequencing depth was reached

We apologise that this data was not included in the previous version, and these details have now been added to the Methods section and shown in the new Supplementary Figure 1.

Reviewer #2: The manuscript ” Changes in the gut microbiota of Nigerian infants within the first year of life” PONE-D-21-33595 by Oyedemi et al. describes microbiota development in the Nigerian population and contributes to filling gaps in our knowledge in the microbiota development trajectories in non-western population. The contribution is in this respect very valuable. The manuscript is in general well written and reported. However, I have only some concerns about the methodologies and some suggestions for the improvement of manuscript, as follows:

Introduction:

Although there no studies on the development of the infant gut microbial composition over time from African countries, there are some cross-sectional studies investigating microbiota in African children. Please, describe the main findings from these studies and how microbiota was found to differ in African vs other populations -in this respect direct comparisons may not be available, but major differences may be obvious based on the literature.

We appreciate the reviewer pointing out that we should expand our comparison to these papers – and this has now been done. We have added additional text to the Introduction and Discussion to put our study in context.

M&M

Sample collection was done by immersing the sample in ethanol for 24 h and then drying. Are there previous studied comparing microbiota results obtained by using this method vs analysing fresh (or immediately -80C frozen) samples. If yes, please provide the reference. If not, and you would have such results, please provide them in supplementary material. The collection method is very practical and useful especially in low income or field settings and it would be good to know to what extent it has been validated. Is the protocol the same that was used in refs 17 and 18 or modified?

We acknowledge the importance of this practical protocol for collecting samples from field settings. We have now added the original reference using this method (Roedar et al 2004), and have provided additional information about the validation carried out in the Schnorr et al Nature communications manuscript (reference 17), which we had previously cited. In the latter paper the authors compared the desiccation method with -80oC storage for divided samples, showing comparable data for DNA yield, gut microbiota and SCFA relative abundance profiles. This has been added to the text. We also validated the method ourselves, illustrating that microbial composition profiles clustered per individual rather than processing method, and have commented on this in the text and added this data to supplemental information.

L115: “0.1g of each faecal sample” -add “dry weight” after 0.1 g

This has been done

L159: The subtitle “Verification of manufacturer’s claim on NAN 1 formula milk” seems like an advertisement and doesn´t well describe the contents of the paragraph. Change to “Detection of bifidobacteria in formula milk by genus-specific PCR”

We appreciate the reviewer pointing this out and have made the change as suggested 

Has the method for measuring short chain fatty acids (SCFAs) been validated for ethanol treated and dried faecal samples? Ref 34, which is given as reference, investigated fresh faecal samples presumably. Concerning SCFA results, it is critical to know the validation as the analysis can be expected to be affected largely by sample preparation. Without validation the results may not be reliable and thereof should not be presented.

We appreciate the review’s concern on this point which reflected our own feelings. Reference 34 indeed investigates the SCFA profile of samples for which initial processing used fresh samples. However the cited reference 17, which utilised the same 2-step desiccation method we used for faecal processing prior to microbial composition analysis, also compared relative SCFA abundance in frozen and desiccated samples. They showed a maximum of 5% variation in the percentage abundance of the major SCFA in faecal samples from two individuals, illustrating remarkable consistency and validity of the data. We fully appreciate that the absolute concentrations of SCFA in samples that have been desiccated is likely to be lower than that found in fresh/frozen samples, but this validation enables us to have confidence that samples treated in the same way can be compared, and that the relative abundance reflects the amounts present in the original samples. We have however decided to replace our original Figure 4 graph which showed the calculated concentration of SCFA in mM (calculated per g of dried faecal sample) with a new graph showing the relative abundance of the major SCFA expressed as % of the total. Although the concentrations we observed were consistent with those presented in other literature, having considered the reviewer’s comment, we feel the percentage data allows us to robustly compare the samples within the study that were all treated the same way. The significant increases in acetate and butyrate, and decrease in succinate, following weaning were observed in both data sets, while the shifts in propionate and lactate concentrations were different, with the % lactate relatively constant in these samples pre/ and post weaning. The replaced data is described in the text.

Results:

L222 and L225: The word “upregulated” may not be the best to describe “had higher abundance”. Revise throughout the manuscript.

We agree with the reviewer that this phrasing was inaccurate and have corrected this throughout. 

Discussion:

L338:What about in comparison to previous findings in African children or other non-western populations?

Extra detail on this point has been added to the discussion, as per response to reviewer 1

Include in the discussion a general overview on the microbiota development in Nigerian infants as compared to other populations, what is similar and what is different in the general trajectories.

Extra detail on this point has been added to the discussion, as per response to reviewer 1

L365: Correct “species of Bifidobacteria” to “species of Bifidobacterium” and genus name in italic font

We apologise for this error and have now corrected this, here and throughout the manuscript

L367: Correct “Bifidobacterial” to “Bifidobacterium” and genus name in italic font

We apologise for this error and have now corrected this 

L392: “Bacteroides” in italic font. Also Prevotella in L.393 and elsewhere. Check throughout the manuscript that genus names are in italic font

We apologise for this error and have now corrected this here and throughout the manuscript

---

## [Decision Letter · Decision Letter 1]

24 Feb 2022

Changes in the gut microbiota of Nigerian infants within the first year of life

PONE-D-21-33595R1

Dear Dr. Scott,

We’re pleased to inform you that your manuscript has been judged scientifically suitable for publication and will be formally accepted for publication once it meets all outstanding technical requirements.

Please make sure that the following points are corrected in the proofs: 1) Consistently writing of 16S rRNA gene (sometimes I see 16S rRNA), 2) Correct use of the degrees Celsius symbol, 3) Making sure that plural names of genera, such as enterococci and clostridia are not written in italics nor capitalized.

Kind regards,

Erwin G Zoetendal, PhD

Academic Editor

PLOS ONE

Additional Editor Comments (optional):

Reviewers' comments:

Reviewer's Responses to Questions

**Comments to the Author**

1. If the authors have adequately addressed your comments raised in a previous round of review and you feel that this manuscript is now acceptable for publication, you may indicate that here to bypass the “Comments to the Author” section, enter your conflict of interest statement in the “Confidential to Editor” section, and submit your "Accept" recommendation.

Reviewer #1: All comments have been addressed

Reviewer #2: All comments have been addressed

2. Is the manuscript technically sound, and do the data support the conclusions?

Reviewer #1: (No Response)

Reviewer #2: Yes

3. Has the statistical analysis been performed appropriately and rigorously? 

Reviewer #1: (No Response)

Reviewer #2: Yes

4. Have the authors made all data underlying the findings in their manuscript fully available?

Reviewer #1: (No Response)

Reviewer #2: Yes

5. Is the manuscript presented in an intelligible fashion and written in standard English?

Reviewer #1: (No Response)

Reviewer #2: Yes

6. Review Comments to the Author

Reviewer #1: (No Response)

Reviewer #2: The authors have addressed all mu questions/concerns adequately. I would like to congratulate the authors for accomplishing the study.

7. PLOS authors have the option to publish the peer review history of their article (what does this mean?). If published, this will include your full peer review and any attached files.

Reviewer #1: No

Reviewer #2: No

---

## [Editor Report · Acceptance letter]

8 Mar 2022

PONE-D-21-33595R1 

Changes in the gut microbiota of Nigerian infants within the first year of life 

Dear Dr. Scott:

I'm pleased to inform you that your manuscript has been deemed suitable for publication in PLOS ONE. Congratulations! Your manuscript is now with our production department. 

Kind regards, 

on behalf of

Dr. Erwin G Zoetendal 

Academic Editor

PLOS ONE